# Procedural Outcome Following Stent-Assisted Coiling for Wide-Necked Aneurysms Using Three Different Stent Models: A Single-Center Experience

**DOI:** 10.3390/jcm11123469

**Published:** 2022-06-16

**Authors:** Catherine Strittmatter, Lukas Meyer, Gabriel Broocks, Maria Alexandrou, Maria Politi, Maria Boutchakova, Andreas Henssler, Marcus Reinges, Andreas Simgen, Panagiotis Papanagiotou, Christian Roth

**Affiliations:** 1Department of Neurosurgery, Hospital Bremen-Mitte, 28205 Bremen, Germany; catherine.strittmatter@gesundheitnord.de (C.S.); andreas.henssler@klinikum-bremen-mitte.de (A.H.); marcus.reinges@gesundheitnord.de (M.R.); 2Department of Diagnostic and Interventional Neuroradiology, Hospital Bremen-Mitte, 28205 Bremen, Germany; maria.alexandrou@klinikum-bremen-mitte.de (M.A.); mariapoliti@hotmail.com (M.P.); maria.boutchakova@klinikum-bremen-mitte.de (M.B.); panagiotis.papanagiotou@klinikum-bremen-mitte.de (P.P.); 3Department of Diagnostic and Interventional Neuroradiology, University Medical Center Hamburg-Eppendorf, 20246 Hamburg, Germany; lu.meyer@uke.de (L.M.); g.broocks@uke.de (G.B.); 4Interventional Radiology Unit, Evangelismos General Hospital, 10676 Athens, Greece; 5Department of Diagnostic and Interventional Neuroradiology, Westpfalz-Klinikum, 67655 Kaiserslautern, Germany; asimgen@westpfalz-klinikum.de; 6Department of Radiology, Areteion University Hospital, National and Kapodistrian University of Athens, 10679 Athens, Greece

**Keywords:** intracranial aneurysms, endovascular treatment, interventional devices

## Abstract

Previous case series have described the safety and efficacy of different stent models for stent-assisted aneurysm coiling (SAC), but comparative analyses of procedural results are limited. This study investigates the procedural outcome and safety of three different stent models (Atlas™, LEO+™ (Baby) and Enterprise™) in the setting of elective SAC treated at a tertiary neuro-endovascular center. We retrospectively reviewed all consecutively treated patients that received endovascular SAC for intracranial aneurysms between 1 July 2013 and 31 March 2020, excluding all emergency angiographies for acute subarachnoid hemorrhage. The primary procedural outcome was the occlusion rate evaluated with the Raymond–Roy occlusion classification (RROC) assessed on digital subtraction angiography (DSA) at 6- and 12-month follow-up. Safety assessment included periprocedural adverse events (i.e., symptomatic ischemic complications, symptomatic intracerebral hemorrhage, iatrogenic perforation, dissection, or aneurysm rupture and in-stent thrombosis) and in-house mortality. Uni- and multivariable logistic regression analyses were performed to identify patient baseline and aneurysm characteristics that were associated with complete aneurysm obliteration at follow-up. A total of 156 patients undergoing endovascular treatment via SAC met the inclusion criteria. The median age was 62 years (IQR, 55–71), and 73.7% (115) of patients were female. At first follow-up (6-month) and last available follow-up (12 and 18 months), complete aneurysm occlusion was observed in 78.3% (90) and 76.9% (102) of patients, respectively. There were no differences regarding the occlusion rates stratified by stent model. Multivariable logistic analysis revealed increasing dome/neck ratio (adjusted odds ratio (aOR), 0.26.; 95% CI, 0.11–0.64; *p* = 0.003), increasing neck size (aOR, 0.70; 95% CI, 0.51–0.96; *p* = 0.027), and female sex (aOR, 4.37; 95% CI, 1.68–11.36; *p* = 0.002) as independently associated with treatment success. This study showed comparable rates of complete long-term aneurysm obliteration and safety following SAC for intracranial aneurysm with three different stent-models highlighting the procedural feasibility of this treatment strategy with currently available stent-models. Increased neck size and a higher dome/neck ratio were independent variables associated with less frequent complete aneurysm obliteration.

## 1. Introduction

Stent-assisted coiling (SAC) is a popular treatment option for both small and complex intracranial aneurysms, as it facilitates coil packing of wide-necked aneurysms that are at risk of coil dislocation and herniation into the parent target vessel, causing potentially periprocedural or delayed ischemic complications [1,2]. Furthermore, SAC allows higher densities of endo-saccular coil embolization and, as a result, higher occlusion and lower recurrence rates due to improved aneurysm thrombosis in comparison to coiling alone, which is not suitable in the majority of wide-necked aneurysm [3].

In the evolution of intracranial stents approved for assisted-coiling, various types of stents have become commercially available and along with them different procedural advantages [4].

We compared and analyzed the procedural results of three different three self-expanding stent models and hypothesized that all three types of stents were efficient for assisted-coiling of wide-necked aneurysms, resulting in high rates of complete aneurysm obliteration at the time of follow-up.

## 2. Materials and Methods

### 2.1. Study Population

We retrospectively analyzed data from all consecutive patients with intracranial aneurysms endovascularly treated between 1 July 2013 and 31 March 2020 at our comprehensive stroke center. All patients receiving SAC were selected and analyzed.

The angiographic results of three self-expanding stents (LEO+(Baby)™, Balt, Montmorency, France; Neuroform Atlas™, Stryker Neurovascular, Fremont, CA, USA; Enterprise™, Codman Neurovascular, Raynham, MA, USA) were analyzed together with patient baseline characteristics and anatomical aneurysm details treated consecutively at our comprehensive stroke center.

The main inclusion criteria for all cases were (1) the diagnosis of an intracranial aneurysm within the anterior or posterior circulation and (2) endovascular treatment in an elective setting (3) via SAC using the LEO+, Enterprise, or Atlas stent models, regardless of previous treatments (i.e., endovascular solely coiling or surgical clipping). There were no exclusion criteria regarding the choice of navigation and delivery catheters. All emergency angiographies were excluded, including patients with acute subarachnoid hemorrhage (SAH) due to intracranial aneurysm rupture.

For all patients meeting the inclusion criteria, baseline and procedural characteristics were collected from medical records and imaging studies. All data were recorded in accordance with the local ethical review board, and informed consent was waived after review (ethics committee of the chamber of Physicians, Bremen, Germany) due to the retrospective study design using fully anonymized data only.

### 2.2. Antiplatelet and Anticoagulation Therapy

All patients, from 7 days to a minimum of 3 days before the procedure patients, received dual antiplatelet therapy primarily with Acetylsalicylic acid (100 mg daily) and Clopidogrel (75 mg daily). All patients were tested using Multiplate^®^ test or in vitro platelet function for adequate response to anti-platelet therapy, and based on the test results, the medication was adjusted. For non-responders, Clopidogrel was changed to Ticagrelor. At the beginning of the intervention, weight-adapted Heparin was administered intra-arterially. After the procedure, all patients were kept lifelong on daily Acetylsalicylic acid and daily Clopidogrel for a minimum of six month. In cases with signs of in-stent stenosis, the patient was kept on dual antiplatelet therapy for another 3 months until the next follow-up.

### 2.3. Endovascular Procedure

All procedures were carried out under general anesthesia using a biplane angiography system (Philips Allura Xper FD20/15, Koninklijke Philips N.V., Amsterdam, The Netherlands). Cerebral vessel access was established using a 6F 088 Neuron MAX Long Sheath 90/4 Straight (Penumbra, Inc., Alameda, CA, USA). Subsequently, a three-dimensional rotational angiography was performed to plan the procedure. In all cases that involved placement of a LEO+(Baby) or Enterprise stent, jailing technique was used: First, a microcatheter system was navigated in the aneurysm; second, another microcatheter system was navigated distal to the aneurysm, and the stent was then partially unsheathed and the aneurysm was coiled. After the aneurysms were fully packed with coils, the stent was completely unsheathed. In the cases in which the Atlas stent was used, the flow diverter was first deployed, and then a microwire/microcatheter system was navigated through the stent into the aneurysm for subsequent coiling.

### 2.4. Stent Models

The use of each stent model depended on the commercial availability and the preference of the interventionalist in charge of the procedure.

The Neuroform Atlas™ (Stryker Neurovascular, Fremont, CA, USA) is a laser cut, nitinol, self-expanding stent system consisting of a hybrid design with closed cells proximally and open cells distally. Accordingly, the proximal part facilitates microcatheter re-crossing, and the distal part allows for better anchoring and stability within the distal landing zone of the artery, as well as vessel wall apposition. The stent includes radiopaque markers at the proximal and distal parts to guarantee visualization under fluoroscopy. For the stent delivery it is mandatory to use the Excelsior SL-10 0.016 microcatheter (Stryker, Fremont, CA, USA), and stents are available in diameters and lengths of 3.0 to 4.5 mm and 15 to 30 mm, respectively [5,6,7].

The LEO+ (Baby) ™ (Balt, Montmorency, France) is a nitinol self-expanding stent system with a woven closed-cell design. The stent contains two helical platinum wires that radiopaque for the entire length of the stent, allowing full visualization of both length and diameter under fluoroscopy. This feature facilitates identification of the stent position and deployment. The LEO+ (Baby)™ can be placed through every 0.017″ microcatheter lumen, and stent sizes are available in diameters and lengths of 2.0 to 5.5 mm and 12 to 75 mm, respectively [6,8,9].

The Enterprise™ (Codman Neurovascular, Raynham, MA, USA) is a self-expandable nitinol stent with a closed-cell design. The ending of the stent contains four radiopaque markers on each side. The delivery system contains a radiopaque marker in the midportion of the stent for better visualization during the deployment process. For the stent delivery it is mandatory to use the 0.021″ Prowler Plus microcatheter (Codman Neurovascular, Raynham, MA, USA). The Enterprise™ stent is available in diameters and length of 4.5 mm and 14 to 37 mm, respectively [6,10].

### 2.5. Aneurysm Details and Procedural Outcome

Aneurysm details were analyzed including the exact location, size, dome/neck ratio, and pre-/post-target arterial vessel diameters in the aneurysm area. All aneurysm details were stratified by stent models. The angiographic outcome was evaluated post-interventionally, and at the time of follow-up, angiography by digital subtraction angiography (DSA) using the Raymond–Roy occlusion classification (RROC) [11]. In this scheme, Class I is defined as complete obliteration, Class II as residual neck, and Class III as residual aneurysm (5). RROC I was defined as the primary angiographic outcome endpoint, and all occlusion rates were compared by utilized stent models. Patient baseline, aneurysm, and procedural characteristics were analyzed for significant associations with complete aneurysm occlusion (RROC 1) at last 6 month (first follow-up) and last available follow-up, including 12- and 18-month follow-ups.

Periprocedural and long-term complications were defined as ischemic events without being clinically symptomatic (any signs of ischemia on follow-up imaging) or ischemic events leading new relevant disabling neurological deficits, occurrence of SAH on post-procedural follow-up imaging or intracerebral hemorrhage (ICH) with regard to clinical symptoms (sICH, defined as an bleeding event that most likely caused neurological deterioration). Furthermore, the occurrence of in-stent thrombosis, intima hyperplasia or stenosis, or thrombosis periprocedurally or at the time of follow-up imaging, including computed tomography angiography and magnetic resonance imaging.

### 2.6. Statistical Methods

Standard descriptive statistics were applied for all data endpoints. Univariable distribution of metric variables was described with median and interquartile range (IQR). Categorical variables were compared with the Chi-squared or Fishers exact test was performed. The Mann–Whitney U test (non-normally distributed data) was used to compare continuous variables. Uni- and multivariable logistic regression analysis was performed to identify variables (patient baseline characteristics and aneurysm details) associated with complete aneurysm occlusion (RROC 1). Variables included in the model were selected based on available literature and clinical relevance [12,13,14,15,16]. Odds ratios (OR) and adjusted OR (aOR) are presented with 95% confidence intervals. *p* values < 0.05 were considered as statistically significant. All statistical analyses were performed in with SPSS version 27.0 (IBM Corporation, Armonk, NY, USA) and Stata 17.0 (StataMP, StataCorp, College Station, TX, USA).

## 3. Results

### 3.1. Study Population

A total of 156 patients met the inclusion criteria and were treated endovascularly via stent-assisted coiling between 1 July 2013 and 31 March 2020 in an elective setting at our comprehensive stroke center. In 82% (128), the aneurysm was detected incidentally, and in 18% (28), patients initially experienced SAH. Overall, 19.9% (31) of cases had been treated previously with solely coiling (18.6%; 29) or surgical clipping (1.3%; 2). Out of these previously treated aneurysms, 87% (27/31) had been initially treated for acute SAH. The median age was 62 years (IQR, 55–71) and 73.7% (115) were female. The most frequent risk factor associated with intracranial aneurysms was arterial hypertension, with 45.5% (71) and in 44.9% (70) multiple aneurysms were detected. Most of the treated aneurysms were located in the anterior circulation (85.3%; 133) with the highest frequencies in the internal carotid artery (20.5% 32), the middle cerebral artery (41.7%; 65), and the anterior communicating artery (19.9%; 31). The most frequent location in the posterior circulation (14.7%; 23) was the basilar artery (10.2%; 16). The mean dome diameter was 5.9 mm (SD ± 3), the mean neck diameter was 4 mm (SD ± 2.1), and the mean dome/neck ratio was 1.5 (SD ± 0.57). The pre-target arterial vessel diameter was 3 mm (SD ± 0.78) and the post-target arterial vessel diameter was 2.4 mm (SD ± 0.67). The median number of implanted coils was 6 (IQR, 4–8). Table 1 provides an overview of patient characteristics stratified by stent model.

### 3.2. Procedural Outcome and Complications

On Raymond–Roy classification, complete obliteration (RROC I) at 6-month follow-up was observed in 78.3% (90) of patients. Stratified by stent model (Figure 1), RROC I was observed in 82.4% (61) with the LEO+ stent, in 71.4% (10) with the Enterprise, and 70.4% (19) with the Atlas stent. At last available follow-up, complete obliteration (RROC I) was observed in 76.9% (102) of patients. Stratified by stent model (Figure 1), RROC I was observed in 83% (69) with the LEO+ stent, in 66.7% (12) with the Enterprise, and 70% (21) with the Atlas stent.

Periprocedurally, two iatrogenic dissections (1.3%; 2) of the internal carotid artery occurred. In five cases (3.2%, 5), a coil dislocation was observed. An in-stent thrombosis was observed in 3.2% (5) of all cases. On postprocedural magnetic resonance imaging follow-up, ischemic-restricted diffusions, without being clinically symptomatic, were present in 16% (25) of cases. Observation of ICH and SAH occurred in one (0.6%, 1) and two cases (1.3%, 2), respectively. sICH occurred in one case (0.6%, 1), leading to in-hospital death. In one case, a new disabling clinical deficit was post-procedurally present (0.6%, 1). In-stent hyperplasia and in-stent stenosis occurred in 5.1% (8) and 5.8% (9), respectively. Table 2 provides an overview of procedure-related complications.

### 3.3. Logistic Regression Analysis for Complete Occlusion

In univariable logistic analysis, female sex (OR, 5.0; 95% CI 2.95–12.16; *p* = 0.001), increasing dome size (OR, 0.75; 95% CI 0.63–0.89; *p* = 0.001), time from onset to groin (OR, 5; 95% CI 2.95–12.16; *p* = 0.001), increasing neck size (OR, 0.74; 95% CI 0.58–0.95; *p* = 0.017), increasing dome/neck ratio (OR, 0.45; 95% CI 0.22–9.23; *p* = 0.029), and increasing pre-target vessel diameter (OR, 0.52; 95% CI 0.29–9.24; *p* = 0.026), were associated factors with complete aneurysm obliteration at last available follow-up (Table 3 including Logistic regression analysis at 6 months follow-up).

Multivariable logistic analysis confirmed increasing dome/neck ratio (adjusted odds ratio (aOR), 0.26.; 95% CI, 0.11–0.64; *p* = 0.003), increasing neck size (aOR, 0.70; 95% CI, 0.51–0.96; *p* = 0.027), and female sex (aOR, 4.37; 95% CI, 1.68–11.36; *p* = 0.002) as independent factors associated with complete aneurysm occlusion at last available follow-up (Figure 2).

## 4. Discussion

Our study on the impact of different stent models on angiographic outcome following SAC for wide-necked aneurysms revealed several findings: (1) a similar high rate of complete obliteration on long-term follow-up can be expected when performing SAC with different stent models (Atlas™, Enterprise™, LEO+™); (2) periprocedural complication rates were relatively low when compared previous studies on SAC; and (3) increased neck size and a higher dome/neck ratio were independent variables associated with less frequent complete aneurysm obliteration.

Wide-necked aneurysms are, in some cases, untreatable or difficult to treat with standard coiling [17]. For these cases, endovascular therapy strategies include flow diverters with and without additional coiling, balloon or temporary stent-assisted (i.e., Comaneci) coiling, and coiling in assistance with intracranial stents. In the latter, the stent covers the neck, stabilizing the coil mass inside the aneurysmal sac and avoids coil herniation into the parent artery. Since the first report of self-expandable stents for intracranial use, multiple advances in stent designs have expanded the commercial availability of different types of intracranial stents for SAC with different properties and, thus, technical advantages and disadvantages [18].

In our case series, we compared three self-expandable nitinol stents, including the Atlas™, with a lase—cut hybrid closed/open-cell design, the Enterprise™, with a laser-cut closed-cell design, and the LEO+(Baby)™, with a woven close-cell design. Additionally, open-cell stent designs have the advantage of improved wall apposition in tortious vessels but the disadvantage that re-sheathing is impossible and the increased cell size, especially in the vertex region of curved vessels, provokes the risk of coil dislocation and subsequent prolapse. Closed-cell stent designs, on the other hand, contain the risk of poor wall apposition and kinking tortious vessel curves. We observed a relatively high rate of complete aneurysm obliteration (RROC I) of 78% in the whole study cohort, which is in line with previous studies investigating SACs ranging from 54–81% with different stent models [4,5,19,20,21]. In our study the rate did not differ between different stent models but was numerically higher in the closed-cell LEO+™ stent group. This finding may be explained by the higher mesh density of the LEO+™ compared with the other stents; however, a bias due to the higher frequency of use of this stent at our department combined with greater experience, which is known to be critical for better outcomes in neuroendovascular procedures, cannot be excluded [22]. Additionally, Pumar et al. observed slightly lower rates (73%) of complete occlusion at long-term follow-up [23]. Multivariable logistic regression analysis showed that an increasing dome/neck ratio and increased neck size were independently associated with incomplete aneurysm obliteration. At first glance, these results seem contradictory, but they must be interpreted in the context of wide-necked aneurysms with relatively increasing neck and aneurysm sizes. Previous studies corroborated that that especially large and wide-necked aneurysms tend to show incomplete aneurysm obliteration at the time of follow-up. Interestingly, even in our cohort that only consisted of wide-necked aneurysms, these findings were stable and may be considered a priori as negative prognostic predictors for complete occlusion in patients treated with SAC for wide-necked aneurysms. Furthermore, female sex was independently associated with aneurysm obliteration, which is most likely biased by the fact that aneurysm cohorts are likely to include higher numbers of women than men because female sex is a common risk factor for developing intracranial aneurysms and rupture [24,25,26].

The endovascular procedure of SAC has been proven to be effective in the past but remains a technical challenging intervention that is associated with increased risks for ischemic events and intracranial hemorrhage due to the necessity of dual antiplatelet therapy prior and after the procedure. Additionally, balloon-assisted coiling may have an advantage regarding the control of medical risk factors for periprocedural hemorrhage but increases the weight of ischemic risk factors. Additionally, the rates of ischemic events and intracranial hemorrhage after SAC in the literature differ around 5% and 2%, respectively [19]. In our case series, we observed low rates of hemorrhagic and ischemic events, highlighting the general safety of the procedure. However, one patient died after a symptomatic bleeding event, and one patient experienced new and disabling deficits leading to a permanent hemiparesis. Nevertheless, these periprocedural complications are marking a devastating clinical course in an elective treatment setting. Important long-term complications after SAC are in-stent intima hyperplasia and stenosis. Our results of an 6% in-stent stenosis rate corroborated previous studies that reported on in-stent thrombosis rate of 10.2% and did observe differences in the risk of developing in-stent stenosis between different stent models such as Atlas and Enterprise [27,28,29]. However, this delayed complication underlines the importance of follow-up imaging not only for detecting aneurysm recurrence but also for cases with in-stent stenosis that possibly need to be re-treated [30].

### Study Limitations

This study has all limitations that are associated with a retrospective single-center study, including missing data (e.g., follow-up imaging) in a certain number of cases and unequal numbers of patients in each stent group. Additionally, occlusion rates were not assessed by an independent core laboratory; however, the treatment results were evaluated by an independent and experienced analyst that did not take part in the procedure. Follow-up data on functional and clinical outcome were not available.

## 5. Conclusions

This study showed comparable rates of complete long-term aneurysm occlusion rates and following SAC for intracranial aneurysm with three different stent-models. Increased neck size and a higher dome/neck ratio were independent variables associated with less frequent complete aneurysm obliteration. Periprocedural complications were comparatively low highlighting the procedural feasibility and safety of this endovascular treatment strategy with currently available stent models.

## Figures and Tables

**Figure 1 jcm-11-03469-f001:**
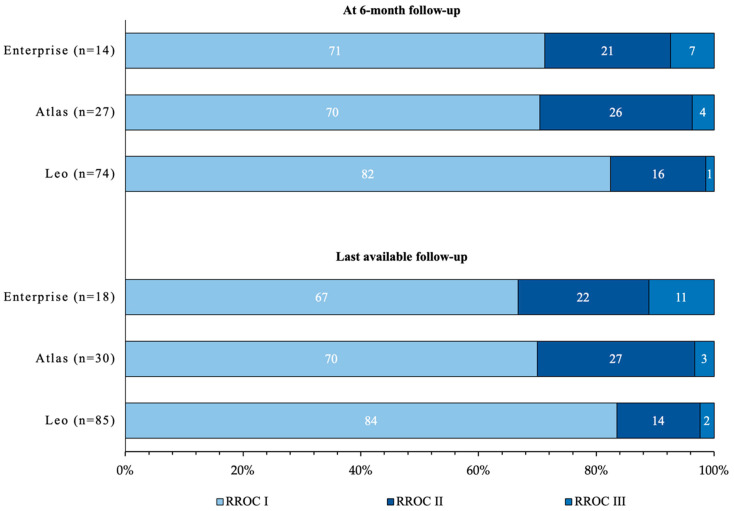
Rates of aneurysm obliteration assessed with Raymond–Roy Occlusion Classification (RROC) stratified by stent model and time of follow-up.

**Figure 2 jcm-11-03469-f002:**
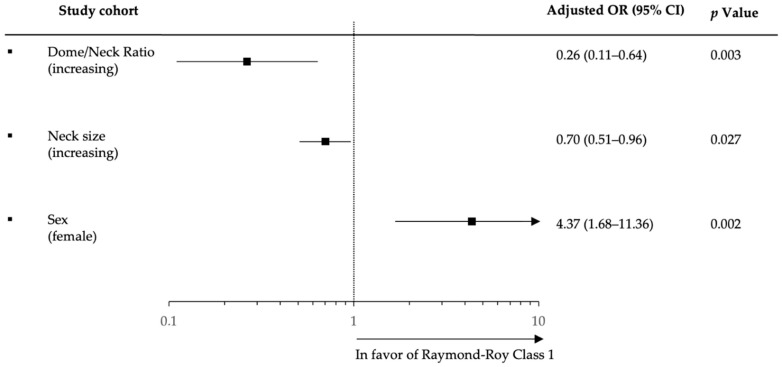
Forest plot based on stepwise multivariable logistic regression analysis for factors associated with complete aneurysm obliteration (Raymond–Roy occlusion class I) at last available follow-up assessed on digital subtraction angiography (DSA).

**Table 1 jcm-11-03469-t001:** Baseline characteristics and anatomical aneurysm details grouped by stent model.

Baseline Characteristics	All Patients	LEO+	Enterprise	Atlas	P1	P2	P3
	(n = 156)	(n = 93)	(n = 19)	(n = 44)			
▪ Age, years; median (IQR)	62 (55–71)	62 (56–73)	59 (48–67)	61 (53.5–68)	0.235	0.276	0.549
▪ Female sex, % (n)	73.7 (115)	72 (67)	73.7 (14)	77.3 (34)	0.884	0.516	0.759
▪ Arterial hypertension, % (n)	45.5 (71)	47.3 (44)	47.4 (9)	59.1 (26)	0.996	0.197	0.390
▪ History of multiple Aneurysms, % (n)	44.9 (70)	51.6 (48)	21.1 (4)	40.9 (18)	0.015 *	0.242	0.129
**Aneurysm detection**, % (n)							
▪ Incidental	82 (128)	75.3 (70)	73.7 (14)	90.9 (40)			
▪ Previously treated	19.9 (31)	-	-	-	-	-	-
- Clipped	1.3 (2)	2.2 (2)	-	-	-	-	-
- Coiled	18.6 (29)	21.5 (20)	26.3 (5)	9.1 (4)	-	-	-
**Aneurysm location**, % (n)							
▪ Anterior circulation	85.3 (133)	86 (80)	94.7 (18)	79.5 (5)	0.457	<0.001	<0.001
- Internal carotid artery	20.5 (32)	19.4 (18)	21.1 (4)	22.7 (10)	-	-	-
- Middle cerebral artery	41.7 (65)	45.2 (42)	31.6 (6)	38.6 (17)	-	-	-
- Anterior communicating artery	19.9 (31)	19.4 (18)	42.1 (8)	11.4 (5)	-	-	-
- Pericallosal artery	3.2 (5)	2.2 (2)	-	6.8 (3)	-	-	-
▪ Posterior circulation	14.7 (23)	14 (13)	5.3 (1)	20.5 (9)	0.457	0.332	0.258
- Vertebral artery	1.9 (3)	3.2 (2)	-	-	-	-	-
- Posterior inferior cerebellar artery	1.9 (3)	3.2 (2)	-	-	-	-	-
- Basilar artery	6.4 (10)	2.2 (2)	-	18.2 (8)	-	-	-
- Basilar tip	3.8 (6)	4.3 (3)	5.3 (1)	2.3 (1)	-	-	-
- Posterior cerebral artery	0.6 (1)	1.1 (1)	-	-	-	-	-
**Aneurysm size**, mm; mean (±SD)							
▪ Dome	5.9 (3)	5.5 (2.7)	5.5 (2.6)	6.7 (3.7)	0.969	0.049	0.109
▪ Neck	4 (2.1)	3.8 (2.1)	3.7 (1.5)	4.6 (2.4)	0.901	0.109	0.214
▪ Dome/Neck ratio	1.5 (0.57)	1.5 (0.54)	1.6 (0.83)	1.5 (0.49)	0.664	0.895	0.626
▪ Vessel diameter pre-target	3 (0.78)	2.9 (0.8)	3 (0.5)	3.1 (0.9)	0.252	0.205	0.851
▪ Vessel diameter post-target	2.4 (0.67)	2.4 (0.7)	2.4 (0.6)	2.5 (0.7)	0.938	0.414	0.559
**Procedural details**, median (IQR)							
Implanted number of coils	6 (4–8)	6 (3–8)	7 (4–9)	7 (4–9)	0.544	0.084	0.741

Legend: * Indicating significance; *p* Values: P1: Leo vs. Enterprise; P2: Leo vs. Atlas; P3: Atlas vs. Enterprise.

**Table 2 jcm-11-03469-t002:** Procedural complications of all patients and grouped by stent model.

Periprocedural Complications	All Patients	LEO+	Enterprise	Atlas
% (n)	(n = 156)	(n = 93)	(n = 19)	(n = 44)
**Technical**				
- Iatrogenic dissection	1.3 (2)	2	-	-
- Coil dislocation	3.2 (5)	2	2	1
**Imaging**				
- Ischemic lesions without clinical deficit *	16 (25)	7	2	16
- In-stent thrombosis	3.2 (5)	4	1	-
- ICH	0.6 (1)	1	-	-
- sICH	0.6 (1)	1	-	-
- SAH	1.3 (2)	-	-	2
- In-stent intima hyperplasia ^#^	5.1 (8)	6	2	-
- In-stent stenosis ^#^	6.8 (9)	7	-	2
**Clinical**				
- New disabling ischemic deficit *	0.6 (1)	1	-	-
- In-hospital Mortality	0.6 (1)	1	-	-

Legend: SAH, subarachnoid hemorrhage; (s)ICH, (symptomatic) intracerebral hemorrhage. * Early in-hospital imaging. ^#^ At the time of follow-up.

**Table 3 jcm-11-03469-t003:** Univariable logistic regression analysis for factors associated with complete aneurysm obliteration (Raymond–Roy occlusion class I) at first and last available follow-up.

	Univariable Logistic Regression Analysis					
	First Follow-Up			Last Available Follow-Up		
	OR	95% CI	*p* Value	OR	95% CI	*p* Value
Age	1.0	0.96–1.04	0.942	0.99	0.9–1.03	0.851
(years)						
Sex	5.9	2.26–15.32	0.001 *	5	2.95–12.16	0.001 *
(female)						
Aneurysm	2.56	0.76–8.68	0.13	1.72	0.55–5.44	0.353
location						
(anterior)						
Leo	1.94	0.79–4.78	0.149	2.24	0.97–5.18	0.059
(yes)						
Atlas	1.76	0.66–4.69	0.259	0.58	0.23–1.45	0.241
(yes)						
Enterprise	0.66	0.19–2.30	0.511	0.51	0.17–1.151	0.224
(yes)						
Dome size (increasing)	0.78	0.66–0.93	0.006 *	0.75	0.63–0.89	0.001 *
Neck size	0.73	0.56–0.95	0.021 *	0.74	0.58–0.95	0.017 *
(increasing)						
Dome/Neck Ratio	0.61	0.29–1.27	0.187	0.45	0.22–9.23	0.029 *
(increasing)						
Vessel diameter pre-target (increasing)	0.41	0.22–0.77	0.006 *	0.52	0.29–9.24	0.026 *
Vessel diameter post-target (increasing)	0.707	0.37–1.35	0.292	0.75	0.41–1.41	0.376

Legend: * Indicating significance; OR = Odds ratio; CI = Confidence Intervals.

## Data Availability

The data that support the findings of this study are available from the corresponding author upon reasonable request.

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
