# Peer review of "Procedural Outcome Following Stent-Assisted Coiling for Wide-Necked Aneurysms Using Three Different Stent Models: A Single-Center Experience"

_jcm, 2022, doi:10.3390/jcm11123469_

Round 1

Reviewer 1 Report

The authors present a single center case series of patients with unruptured cerebral saccular aneurysms treated with sent-assisted coiling, comparing procedural outcome with the use of 3 different stent models (Atlas, LEO Baby and Enterprise). Their results showed comparable occlusion rates.

While the authors should be complimented for the results, the article adds very few to the existing extensive literature on the topic. The implied nuances of the single center comparative study about the different stent models which the authors are aiming to require a thorough analytical method, which this paper is lacking. English language require some editing.

Some remarks:

In the Affiliation list, nr. 1 and 4 are the same.

Abstract:

"comparative analyses of procedural results are sparse". Do you mean "scarse"?

"at 6 months and last available follow-up complete aneurysm occlusion was observed in 78.3 and 77.9%, respectively?" To which stent model are the authors referring? there is a third stent model missing here."

In the Results/Discussion:

Increased neck size and higher dome:neck ratio were associated with less complete aneurysm occlusion rate. If this had a significance, one should expect that lower dome:neck ratio would be associate with lower occlusion rate (since the larger the neck, the lower is the ratio). You should discuss this aspect more. Furthermore, the whole point of performing a stent-assisted coiling is to eliminate the problem of large necks leading to lower occlusion rate. How do the author explain their results?

Introduction:

"SAC allows higher occlusion rate compared to coiling alone who's treatment indication is limited by the aneurysm morphology itself". I guess the authors mean that larger neck aneurysms are not suitable for simple coiling. This paragraph should be rephrased.

"We compared and analyzed..three difference stents.." this paragraph belongs to the Material and Methods section.

"We hypothesized that all three types of stents are efficient.." on which literature do you base this hypothesis? The authors should also discuss alternative coiling methods for unruptured large neck aneurysms: balloon assisted, double-catheter technique and (coil assisted)flow diversion.

Material and Methods:

"All emergency angiographies were excluded including patients with acute SAH". This needs to be reported also in the abstract and simply rephrased as "we included only unruptured saccular aneurysms in our analysis"

The Nr. of the ethic review board should be included.

"At least 7 days before procedure patients received dual antiplatelet therapy" . Which were the exception (longer than 7 days)? How did you decide which patients had to the Clopidogrel longer than 6 months? Did you use Ticagrelor instead of Clopidogrel for any patients?

Which coiling technique was used for the Atlas stents?

On which basis was the use of one or the other stent decided? I assumed it was the year of intervention...please specify.

Study cohort: This was obviously wrongly copied and pasted from the wrong patients population and refers to patients with ischemic stroke treated with IAT". Secondly, it does not belong to this section but rather to the Results.

Follow-up: the follow-up length is not clearly indicated. In the abstract it is 6 to 12 months, in the Methods op to 18 months. please specify

For such a paper primary procedural outcome should also include clinical outcome at follow up such as mRS. In the procedural analysis, wall apposition rate of the stents and coil density score (rather than number of coils, as this latter is highly dependent on the aneurysm size) are important results to be included.

"Statistical analysis": "Variables include din the model were selected based on available literature": please add Citations.

Results:

"Overall, 19.9% of cases had been treated previously with solely coiling or clipping": how many of them were initially ruptured?

"Logistic regression: initial female sex" (remove "initial"), "time from onset to groin" (not relevant as all aneurysms were unruptured and the paper is not about IAT). Hard to explain why female sex should be associated with higher occlusion rate. Does it maybe depend on the fact that more than 70% of the population was female? there are statistical tools to adjust this result.

"Increasing dome size and increasing neck size were associated with completed aneurysm occlusion at 12 months": this is not the same as reported in the Abstract. In Table 3 increasing dome size, increasing neck size and increasing dome:neck ratio were significantly associated with increased occlusion rate. There is something not logical and contradictory  about the whole analysis..maybe the whole analysis does not reach a statistical significance? Could the authors identify a cut-off value for their aneurysm dome, neck, vessels diameters associated with higher occlusion rate? another reported factor in the literature associated with lower occlusion rate is the presence of a persistent fetal circulation for Pcom aneurysms.

Table 2: Procedural complications are not grouped by stent model in this table.

Table 3: Enterprise stent is missing.  

Since the first follow up was at 6 months, there should be two tables, one with the first follow data and another one with the data at the last follow up (12 or 18 months), just like in the bar chart in Fig. 1. 

Fig. 1: in the Y-Axis it seems like the number of patients for each stent (n) between the two follow ups constantly increased. Please explain. Furthermore the %of patients with RROC 1 with the Enterprise stent decreased at the last follow up. What happened to those patients? Did they undergo another coiling?

There is a big statistical difference in the % of patients with RROC1 at last follow up with the three stent (67% vs 70% vs 84%). From your data the occlusion rate with the Leo stent seems way higher than with the other stents. Does it have to do with the much larger cohort of patients or with the operators' learning curve?

Fig. 2: The multivariable logistic regression analysis shows that only female sex is a favorable factor for complete occlusion (again increasing dome:neck ratio or increasing neck ratio - obviously contradicting each other and once again contradicting everything that has been previously discussed in the paper - are risk factors for aneurysm occlusion.

Discussion: Study limitation: which are the limitations associated with a retrospective single-center study? why weren't occlusion rate assessed by an indipendent analyst?

Author Response

We are grateful for the thoughtful comments and suggestions which have improved the manuscript. We indicate below how each comment was addressed (*).

Reviewer 1

- The authors present a single center case series of patients with unruptured cerebral saccular aneurysms treated with sent-assisted coiling, comparing procedural outcome with the use of 3 different stent models (Atlas, LEO Baby and Enterprise). Their results showed comparable occlusion rates.

While the authors should be complimented for the results, the article adds very few to the existing extensive literature on the topic. The implied nuances of the single center comparative study about the different stent models which the authors are aiming to require a thorough analytical method, which this paper is lacking. English language require some editing.

- Some remarks:

1) In the Affiliation list, nr. 1 and 4 are the same.

*Thank you for highlighting this mistake that we corrected.

Abstract: 

2) - "comparative analyses of procedural results are sparse". Do you mean "scarse"?

*With this phrase we wanted to emphasize that the detailed "comparative analyses of procedural results" from different stents is limited. We changed the wording stating now “limited”.

3) "at 6 months and last available follow-up complete aneurysm occlusion was observed in 78.3 and 77.9%, respectively?" To which stent model are the authors referring? there is a third stent model missing here."

In the Results/Discussion:

4) - Increased neck size and higher dome:neck ratio were associated with less complete aneurysm occlusion rate. If this had a significance, one should expect that lower dome:neck ratio would be associate with lower occlusion rate (since the larger the neck, the lower is the ratio). You should discuss this aspect more. Furthermore, the whole point of performing a stent-assisted coiling is to eliminate the problem of large necks leading to lower occlusion rate. How do the author explain their results?

 *We thank the reviewer for this comment and the chance to further explain these findings. In a cohort of per se wide-necked aneurysms (median 4mm), increased neck size was associated with reduced rates of complete aneurysm occlusion. This finding seems to be logical. Contradictory may seem at first glance, that a higher dome/neck ratio was also associated with a reduced rate of complete aneurysm occlusion. However, that does not mean that in these cases with a larger dome the neck itself was small because the whole aneurysm cohort had rather wide necks and therefore, it was a relative increase of the ratio. The fact that both variables were significant in a multivariable model shows that on the one hand, increased neck size (in a cohort of wide-neck aneurysms) and the other hand, increased aneurysm size lead to less frequent rates of complete aneurysm occlusion. Thus, both variables seem to play an important role for prediction of complete aneurysm occlusion in the subgroup of wide necked aneurysms.

Please see now under the section discussion:

At first glance, these results seem contradictory, but they must be interpreted in the context of wide-necked aneurysms with relatively increasing neck and aneurysm sizes.”

Introduction:

5) - "SAC allows higher occlusion rate compared to coiling alone who's treatment indication is limited by the aneurysm morphology itself". I guess the authors mean that larger neck aneurysms are not suitable for simple coiling. This paragraph should be rephrased.

*Thank you for highlighting this incorrect wording that we rephrased as suggested.

Please see under the section introduction:

Furthermore, SAC allows higher densities of endo-saccular coil embolization and by that higher occlusion and lower recurrence rates due to improved aneurysm thrombosis in comparison to coiling alone that is not suitable in the majority of wide-necked aneurysm.”

6) - "We compared and analyzed..three difference stents.." this paragraph belongs to the Material and Methods section.

*Thank you for pointing this out. We shifted this paragraph to the “Material and Methods section” as suggested.

Please see under the section introduction:

“We compared and analyzed the procedural results of three different stent models (LEO+(Baby)™, Balt, Montmorency, France; Neuroform Atlas™, Stryker Neurovascular, Fremont, California, USA; Enterprise™, Codman Neurovascular, Raynham, MA, USA) and hypothesized that all three types of stents are efficient for assisted-coiling of wide-necked aneurysms resulting in high rates of complete aneurysm obliteration at the time of follow-up.”

Please see under the section material and methods section:

The angiographic results of three self-expanding stents (LEO+(Baby)™, Balt, Mont-morency, France; Neuroform Atlas™, Stryker Neurovascular, Fremont, California, USA; Enterprise™, Codman Neurovascular, Raynham, MA, USA) together with patient baseline characteristics and anatomical aneurysm details treated consecutively at our comprehensive stroke center between 07/2013 and 03/2020.”

7) - "We hypothesized that all three types of stents are efficient.." on which literature do you base this hypothesis? The authors should also discuss alternative coiling methods for unruptured large neck aneurysms: balloon assisted, double-catheter technique and (coil assisted)flow diversion.

*Thank you for this comment. This hypothesis derived from our clinical practice and experience. Since all stents have been approved as medical devices and considered to be “effective” for the treatment of aneurysms we wanted to compare them. Therefore, we formulated the hypothesis and analyzed our data in order to investigate it. As suggested, we further discussed alternative treatment options in the discussion section.

Material and Methods:

8) - "All emergency angiographies were excluded including patients with acute SAH". This needs to be reported also in the abstract and simply rephrased as "we included only unruptured saccular aneurysms in our analysis"

*Thank you for highlighting this missing fact. Accordingly, we added this information to the abstract.

Please see under the section abstract:

“We retrospectively reviewed all consecutively treated patients that received endovascular SAC for intracranial aneurysms between 07/2013 and 03/2020 excluding all emergency angiographies for acute subarachnoid hemorrhage.”

9) - The Nr. of the ethic review board should be included.

*Thank you. As suggested, we included the reference number of the ethic review board and their acceptance for analyzing retrospective fully anonymized data in the methods section.

10) - "at least 7-days before procedure patients received dual antiplatelet therapy" . Which were the exception (longer than 7 days)? How did you decide which patients had to the Clopidogrel longer than 6 months? Did you use Ticagrelor instead of Clopidogrel for any patients?

*Thank you for the comment. We evaluated all cases again and patients received from 7 to a minimum of 3 days before the procedure dual antiplatelet therapy. For patients that did not respond to the medication we indeed changed Clopidogrel to Ticagrelor. Only patients that showed signs of in-stent stenosis were kept on dual antiplatelet therapy longer than 6 months and were scheduled for closer follow-up. We adjusted this paragraph accordingly.

Please under methods section:

“All patients were tested using Multiplate® test or in vitro platelet function for adequate response to anti-platelet therapy and based on the test results the medication was adjusted and for non-responders, Clopidogrel was changed to Ticagrelor.” … “In cases with sings of in-stent stenosis, the patient was kept on dual antiplatelet therapy for another 3 months until the next follow-up.”

11) - Which coiling technique was used for the Atlas stents?

*Thank you for mentioning this point. Due to the higher porosity of the Atlas stent, the flow diverter was first deployed and then a microwire/microcatheter-system was passed through the stent in the aneurysm for subsequent coiling. We adjusted this section accordingly.

Please under methods section:

In the cases in which the Atlas stent was used, the flow diverter was first deployed and then a microwire/microcatheter system was inserted through the stent into the aneurysm for subsequent coiling.”

12) - On which basis was the use of one or the other stent decided? I assumed it was the year of intervention...please specify.

*Thank you for this comment. Indeed, the use of each stent depended on the commercial availability The use of each stent model depended on the commercial availability and the preference of the interventionalist in charge of the procedure. We added this information to the material and methods section.

Please under methods section:

The use of each stent model depended on the commercial availability and the preference of the interventionalist in charge of the procedure.”

13) - Study cohort: This was obviously wrongly copied and pasted from the wrong patients population and refers to patients with ischemic stroke treated with IAT". Secondly, it does not belong to this section but rather to the Results.

*We apologize for this mistake that occurred while using a previously submitted JCM template. We deleted this section accordingly.

14) - Follow-up: the follow-up length is not clearly indicated. In the abstract it is 6 to 12 months, in the Methods op to 18 months. please specify

*Thank you. We corrected this incongruency in the abstract and methods section, stating now that the first follow-up was performed after 6 month and the last available follow-up included data of the follow-ups at 12 and 18 months after the procedure.

15) - For such a paper primary procedural outcome should also include clinical outcome at follow up such as mRS. In the procedural analysis, wall apposition rate of the stents and coil density score (rather than number of coils, as this latter is highly dependent on the aneurysm size) are important results to be included.

*Thank you for this comment. We agree that it would be favorable to include clinical outcomes, but this study focuses on the procedural aspects and therefore, this analysis lies outside of the scope of the study. Even though it is complicated to quantify the wall apposition rate and coil density, it would be interesting to analysis these variables. Unfortunately, we cannot provide these parameters due to the retrospective nature of the study.

"Statistical analysis": "Variables include din the model were selected based on available literature": please add Citations.

*Thank you for this comment. As suggested, we referenced the relevant literature for the variable selection.

Results:

16) - "Overall, 19.9% of cases had been treated previously with solely coiling or clipping": how many of them were initially ruptured?

*Thank you for bringing this missing information up. As suggested, we added the information to the results section.

Please see under results section:

Out of these previously treated aneurysms, 87% (27/31) had been initially treated for acute subarachnoid hemorrhage.”

17) - "Logistic regression: initial female sex" (remove "initial"), "time from onset to groin" (not relevant as all aneurysms were unruptured and the paper is not about IAT). Hard to explain why female sex should be associated with higher occlusion rate. Does it maybe depend on the fact that more than 70% of the population was female? there are statistical tools to adjust this result."Increasing dome size and increasing neck size were associated with completed aneurysm occlusion at 12 months": this is not the same as reported in the Abstract. In Table 3 increasing dome size, increasing neck size and increasing dome:neck ratio were significantly associated with increased occlusion rate. There is something not logical and contradictory  about the whole analysis..maybe the whole analysis does not reach a statistical significance? 

*Thank you for this comment. As suggested, we deleted the wrong words (“initial/“onset-to-groin”). Indeed, it seems that the variable “female” was associated with complete aneurysm occlusion due to the higher frequency in the cohort itself and we included this finding in the discussion section. With reference to comment 4, we would like to highlight again that the variables dome/neck ratio and neck have to be interpreted in the light of a wide neck aneurysms. If the neck is wide but the aneurysm is very large than the ratio increases and therefore, the variable “dome” was also significant in univariable logistic regression testing. However, in the multivariable model the variable “dome” did not remain significant, most likely due to the collinearity with the variable “dome/neck” ratio.

19) -Could the authors identify a cut-off value for their aneurysm dome, neck, vessels diameters associated with higher occlusion rate? another reported factor in the literature associated with lower occlusion rate is the presence of a persistent fetal circulation for Pcom aneurysms.

*Thank you for this interesting thought. We performed further statistical tests (e.g., ROC analysis and graphical comparisons) but were unable to identify specific cut-off values for the vascular/aneurysm parameters, which is likely due to the need for a larger data set than that analyzed in this study. Unfortunately, aneurysm branching data were not collected for this study and therefore cannot be provided.

20) -Table 2: Procedural complications are not grouped by stent model in this table.

*We agree with the reviewer that for the sake of stringency this table should also be grouped by stent model even though the cases are numerically very low. As suggested, we adjusted the table.

21) -Table 3: Enterprise stent is missing.

*Thank you for highlighting this mistake. We added the results of the univariable logistic regression analysis for the Enterprise stent in the table.

22) - Since the first follow up was at 6 months, there should be two tables, one with the first follow data and another one with the data at the last follow up (12 or 18 months), just like in the bar chart in Fig. 1. 

*Thank you for this suggestion. As suggested, we added to table 2 the values for all variables of the logistic regression analysis at the 6-month follow-up.

23) - Fig. 1: in the Y-Axis it seems like the number of patients for each stent (n) between the two follow ups constantly increased. Please explain. Furthermore the %of patients with RROC 1 with the Enterprise stent decreased at the last follow up. What happened to those patients? Did they undergo another coiling?

*Thank you for this comment. The increase in follow-ups is explained by inconsistent compliance of some patients that did not want do have a follow-up DSA after 6 month and changed their mind and came after 12 months. Also, the second “last available follow-up” contains data after 12 and 18 month and therefore the number of follow-up cases is higher in the second part of the bar graph.

24) -There is a big statistical difference in the % of patients with RROC1 at last follow up with the three stent (67% vs 70% vs 84%). From your data the occlusion rate with the Leo stent seems way higher than with the other stents. Does it have to do with the much larger cohort of patients or with the operators' learning curve?

*Thank you for highlighting this finding. We agree that there is a certain difference in complete occlusion rates, even though this difference was not significant. This finding was also highlighted by reviewer 2 and we discussed possible explantations in more detail in the discussion section. This finding could be partially explained by the higher mesh density of the LEO stent compared to the other stents leading to less blood flow in the aneurysm and increased thrombosis with subsequent higher occlusion rates.

Please see under discussion section:

This finding may be explained by the higher mesh density of the LEO+™ compared with the other stents; however, a bias due to the higher frequency of use of this stent at our department combined with greater experience, which is known to be critical for better outcomes in neuroendovascular procedures, cannot be excluded."

25) - Fig. 2: The multivariable logistic regression analysis shows that only female sex is a favorable factor for complete occlusion (again increasing dome:neck ratio or increasing neck ratio - obviously contradicting each other and once again contradicting everything that has been previously discussed in the paper - are risk factors for aneurysm occlusion.

*Thank you for this comment. Again, we would like to explain these findings are not contradictory at all in the light of wide-necked aneurysms. An increase in the dome/neck ratio does in wide-necked aneurysms does not mean per se that the neck decreases. These aneurysms simply had a different configuration with a wide-neck and an even larger dome. In the multivariable model, these variables were independently associated with aneurysm occlusion in a negative way with a lower chance (odds ratio <1 ) for complete occlusion.

26) - Discussion: Study limitation: which are the limitations associated with a retrospective single-center study? why weren't occlusion rate assessed by an indipendent analyst?

*Thank you for this comment. The main limitations of our retrospective study were that a certain number of patients dropped out of the follow-up program leading to missing data, a lack of data regarding some variables of interest for further procedural analysis. The data was not analyzed by an independent core lab but an experienced colleague that did not take in the procedure. We corrected this information in the limitation section.

Please see under limitation section:

“This study has all limitations that are associated with a retrospective single-center study in-cluding missing data (e.g., follow-up imaging) in a certain number of cases and unequal numbers of patients in each stent group. Additionally, occlusion rates were not assessed by an independent core laboratory; however, the treatment results were evaluated by an independent analyst that did not take part in the procedure.” Follow-up data on functional and clinical outcome were not available.”

Reviewer 2 Report

The content is helpful, perhaps can explain more why the Leo stent had higher occlusion rates viz a viz its device properties.  

One glaring editing mistake in this paper is that under Study Cohort  (2.5), the authors describe a cohort of stroke patients and is likely describing another paper/study.

Author Response

Reviewer 2

The content is helpful, perhaps can explain more why the Leo stent had higher occlusion rates viz a viz its device properties.  

*Thank you for this comment. Indeed, this finding may be explained by the higher mesh density compared to other stents. However, there may be a partial bias based on the higher frequency this stent was utilized in our center. Accordingly, we added a paragraph on this matter to the discussion section.

Please see under the section discussion:

"This finding may be explained by the higher mesh density of the LEO stent compared with the other stents; however, a bias due to the higher frequency of use of this stent at our department combined with greater experience, which is known to be critical for better outcomes in neuroendovascular procedures, cannot be excluded."

One glaring editing mistake in this paper is that under Study Cohort  (2.5), the authors describe a cohort of stroke patients and is likely describing another paper/study.

*We apologize for this mistake that occurred while using a previously submitted JCM template. We deleted this section accordingly.

Round 2

Reviewer 1 Report

The authors revised the manuscript according to the reviewers’ comments.